# SuperMask: Generating High-resolution object masks from multi-view, unaligned low-resolution MRIs

**Hanxue Gu** [1]                       HANXUE.GU@DUKE.EDU
**Hongyu He** [1]                       HONGYU.HE@DUKE.EDU
**Roy Colglazier** [2]                  ROY.COLGLAZIER@DUKE.EDU
**Jordan Axelrod** [3]                 JORDAN.AXELROD@DUKE.EDU
**Robert French**[2]                  ROBERT.FRENCH@DUKE.EDU
**Maciej A Mazurowski**[*1,2,3,4]        MACIEJ.MAZUROWSKI@DUKE.EDU

[1] *Department of Electrical and Engineering Department, Duke University, NC, USA*

[2] *Department of Radiology, Duke University, NC, USA*

[3] *Department of Computer Science, Duke University, NC, USA*

[4] *Department of Biostatistics and Bioinformatics, Duke University, NC, USA*

**Editors:** Accepted for publication at MIDL 2023

## Abstract

Three-dimensional segmentation in magnetic resonance images (MRI), which reflects the true shape of the objects, is challenging since high-resolution isotropic MRIs are rare and typical MRIs are anisotropic, with the out-of-plane dimension having a much lower resolution. A potential remedy to this issue lies in the fact that often multiple sequences are acquired on different planes. However, in practice, these sequences are not orthogonal to each other, limiting the applicability of many previous solutions to reconstruct higher-resolution images from multiple lower-resolution ones. We propose a weakly-supervised deep learning-based solution to generating high-resolution masks from multiple low-resolution images. Our method combines segmentation and unsupervised registration networks by introducing two new regularizations to make registration and segmentation reinforce each other. Finally, we introduce a multi-view fusion method to generate high-resolution target object masks. The experimental results on two datasets show the superiority of our methods. Importantly, the advantage of not using high-resolution images in the training process makes our method applicable to a wide variety of MRI segmentation tasks. The code for reproducing the results is available at: https://github.com/mazurowski-lab/Supermask.

**Keywords:** High-resolution object generation, medical image segmentation

## 1. Introduction

As a non-invasive and low-radiation imaging technique, magnetic resonance imaging (MRI) plays an important role in disease diagnosis and characterization. In clinical practice, MRI scans are done with relatively few slices and a significant slice thickness, owing to the limits imposed by the technique's slow acquisition speed. In replace of an isotropic high-resolution 3D volume, highly anisotropic images, which can be seen as a stack of 2D slices (Erasmus et al., 2004), are acquired with better resolution inside the slices than in the slice-selection (or through-plane) direction (Van Reeth et al., 2012), see Figure 1 (green box). When directly segmenting the target objects on the anisotropic volumes, the low resolution (LR)

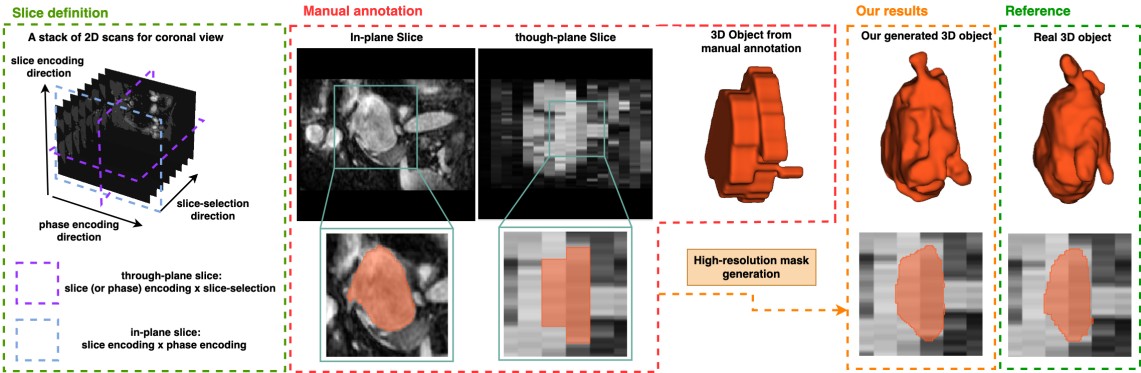

Figure 1: The objective of our strategy is to generate realistic 3D masks for non-isotropic MRIs (have low-resolution at through-plane directions). The left is an illustration of the definition of in-plane slices and through-plane slices. The 3D object directly from manual annotations has a coarse and grid-like appearance (the first 3D object); our method is targeted to generate higher-detailed and more realistic high-resolution masks (the middle 3D object) when only low-resolution images and annotations are available. The generated 3D object has more details and higher similarity to real objects (the right 3D object).

in the slice-selection direction might result in extremely coarse predictions (Figure 1, red box) that are far from the real-object shape, hence hindering the later diagnosis.

Machine learning-based super-resolution (SR) methods have been widely used for the reconstruction of a high-resolution (HR) magnetic resonance (MR) 3D volume from multi-planar LR 2D scans (Van Reeth et al., 2012; Plenge et al., 2012; Gholipour et al., 2010; Jia et al., 2017; Sui et al., 2019). In recent years, several studies have further investigated convolutional neural networks (CNNs) based architectures for HR MRI reconstruction (Pham et al., 2019; Jurek et al., 2020). Ebner et al. (2020) presented a fully automated framework for fetal brain reconstruction that includes coarse fetal brain localization, fine segmentation, and super-resolution reconstruction. Chai et al. (2020) used a generative adversarial network (GAN) to restore the through-plane slices. However, the majority of the previously proposed methods are impractical because current MRI high-resolution recreation methods from multi-planar views often require **well-aligned** and near-perfect orthogonal images (Zhang et al., 2021; Zhou et al., 2019). This is not commonly happening in the real case because the views are defined in the anatomy coordination instead of the world coordination, and patients may switch poses largely between views, as seen in Appendix A. Images can be largely misaligned because of the patient's movement, or other views are not taken simultaneously.

When the resolution at the through-plane direction is too low to distinguish the details, it is impossible to extract precise and realistic masks from a single MRI volume, as seen in Figure 1 (left volume), and applying multi-planar scans into segmentation tasks is also an effective strategy. In order for the recreation of HR masks from multi-view MRIs, image

registration and alignment are essential. Some studies (Askin Incebacak et al., 2022) mentioned an initial registration procedure as the preprocessing. However, traditional registration techniques (Ferrante and Paragios, 2017) are time-consuming, isolated to downstream tasks, and considered different views are taken simultaneously with only small variations in the patient's mobility instead of substantial posture switching. On the other side, our target images have larger displacements, making the registration significantly harder.

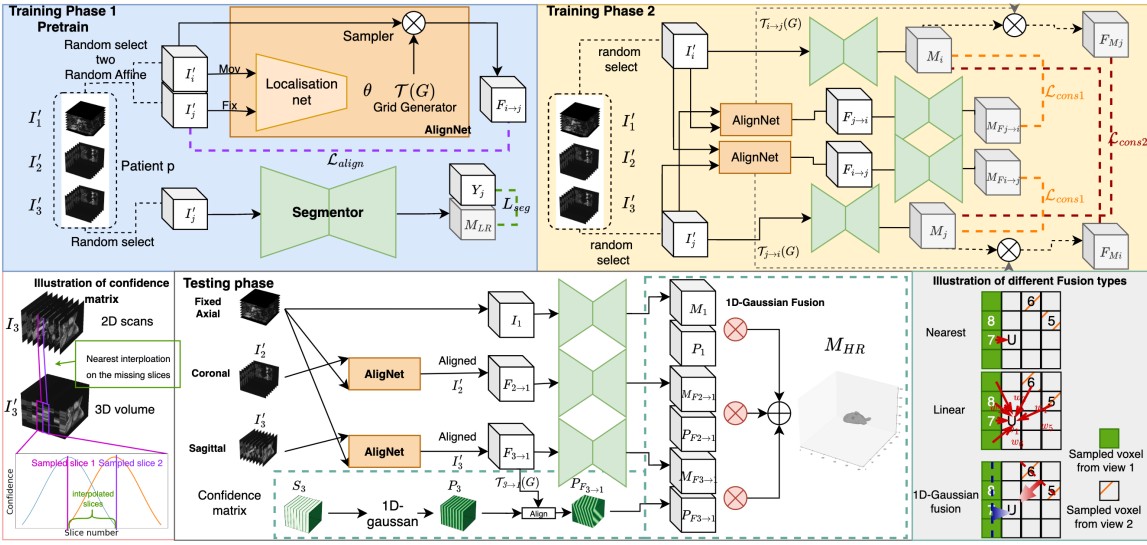

Figure 2: This is the pipeline of our proposed method SuperMask. In training phase 1, an image registration network (AlignNet) and a segmentation network are trained separately. In training phase 2, the AlignNet and Segmentor are intertwined to fine-tune with two newly introduced regularization losses. In the test phase for our method, Coronal and Sagittal view images are aligned into the axial view first, and a 1-D Gaussian fusion is applied to generate a high-resolution mask.

In this paper, we introduce a weakly-supervised framework called *SuperMask* that can automatically register low-resolution, unaligned 2D MRI scans obtained from different orientations and produce HR and precise masks. Our approach utilizes deep learning techniques to learn the deformations required to register the images and the segmentations of target subjects on different views of LR scans. The framework consists of three stages: in the first, registration and segmentation networks are pre-trained independently; in the second, segmentation and registration mutually promote one another (referred to as intertwined learning); and in the third, segmentation results are fused. By introducing two extra regularization terms, our innovative design of intertwined learning can facilitate each part's learning. This complementary learning approach can achieve an optimal registration and target object segmentation solution. In addition, our simple single-dimensional uncertainty Gaussian fusion, inspired by Gaussian Mixture Models (GMM) (Reynolds, 2009), can efficiently fuse information from multiple views, and we consider it a good fit for MRIs' through-plane uncertainty compared with other fusion strategies.

We demonstrate the effectiveness of our framework on a variety of datasets with varying distances between slices. The experiments show that our approach is able to outperform previous methods, particularly in cases where the images are low-resolution or poorly aligned. In contrast with previously generative-based methods (Chai et al., 2020; Yuan et al., 2020) and image-to-image translation methods (Masutani et al., 2020), our method **does not require any high-resolution images or masks** to be involved in the training and design processes. Since high-resolution MRI images are exceedingly rare for several body parts (3D), our method is extensible and effective for a variety of use cases.

## 2. Method

Our *SuperMask* is made to effectively auto-register low-resolution (LR) and unaligned images, segment the targets, and generate HR masks that have more accurate target objects' shape representations. Section 2.2 describes a coarse pre-training for image registration and segmentation, followed by an intertwined training for fine-tuning in Section 2.3. Finally, we show the generation of HR masks using our novel 1D-Gaussian fusion in Section 2.4.

### 2.1. Pre-processing

In the target task, each patient contains three low-resolution 2D MR scans from different views (axial, coronal, and sagittal) of the same body part, $I_1 \in \mathbf{R}^{X \times Y \times Z_l}$, $I_2 \in \mathbf{R}^{X \times Y_l \times Z}$, $I_3 \in \mathbf{R}^{X_l \times Y \times Z}$. $X_l, Y_l, Z_l$ represent the number of slices on lower-sampled dimensions, which are much smaller than the in-plane dimensions $X$, $Y$, and $Z$. The ground-truth masks for these three images are $Y_1, Y_2, Y_3$ of the same dimension. For 2D MRI, which is often not isotropic, its voxel size can be expressed in terms of pixel spacing (p) and distance between slices (d). For example, $I_1$ has a voxel size of $p \times p \times d, d_s = \frac{p}{d} = \frac{Z_l}{Z}$, where we define $d_s$ as a relative ratio of the slice distance vs. in-plane pixel spacing. In order to better register $I_1$, $I_2$, and $I_3$, we first need to match their voxels to the same unit, i.e., by up-sampling and fill the missing slices by nearest interpolation on 2D scans and masks to get a voxel size of $p \times p \times p$, seen as Figure 2 (bottom left). We get 3D volumes $I_1', I_2', I_3', Y_1', Y_2', Y_3' \in \mathbf{R}^{X \times Y \times Z}$.

### 2.2. Coarse segmentation and registration

In *phase 1* training, we train the coarse segmentation and registration separately, as shown in the top left of figure Figure 2. For segmentation, we assemble all of the available up-sampled images and masks into a training set $\mathbf{I'} = \mathbf{I_1'} \cup \mathbf{I_2'} \cup \mathbf{I_3'}$, $\mathbf{Y} = \mathbf{Y_1'} \cup \mathbf{Y_2'} \cup \mathbf{Y_3'}$ and feed them into a 3D U-net (Çiçek et al., 2016) to train a coarse segmentation network $\mathcal{S}_1$. Let $I_i'$ as the input volume, $M_i = \mathcal{S}_1(I_i')$ as the predicted mask and $Y_i$ as the corresponding ground truth mask. Dice loss (Sudre et al., 2017) (noted as $L_{seg_1}$) is applied for penalizing the segmentation objective.

For registration, we implement a coarse 3D spatial transformer network $\mathcal{G}_1$ ((Jaderberg et al., 2015) modified from a 2D version). For each patient, we sample two views, $I_i', I_j' \in I_1', I_2', I_3'$ and apply a random affine transformation to each to augment the training set. The localization network takes two 3D volumes, $I_i'$ as the moving image and $I_j'$ as the fixed image. It generates 12 affine parameters that reflect the transformation from $I_1'$ to $I_2'$.

Considering that MRI responds to real-life objects and does not have the deformation and scaling of objects, we reduce the number of affine parameters by setting the shearing and scaling parameters all to 1. After sampling grid $G$ based on these affine parameters $\theta_{i \to j}$ to get registration field $\phi_{i \to j} = \mathcal{T}_\theta(G)$, we apply a differentiable image sampling on the input moving image $I_i'$ to generate a moved image $f_{i \to j} = \mathcal{G}_1(I_i', I_j') = I_i' \circ \phi_{i \to j}$ that is registered into the pose of $I_j'$.

The unsupervised registration loss combines two components: $\mathcal{L}_{sim}$ that penalizes the difference between the moving and fixed image, and $\mathcal{L}_{ID}$ that prevents the transformation between two identical inputs,

$$\mathcal{L}_{align_1} = \mathcal{L}_{sim}(I_j', f_{i \to j}) + \lambda_1 \mathcal{L}_{ID}, \tag{1}$$

where we apply $L_{sim}$ as local cross-correlation (Boyd, 2001), which is more robust to intensity variations across scans and datasets, and $L_{ID} = ||I_i', \mathcal{G}(I_i', I_i')||_2 + ||I_j', \mathcal{G}(I_j', I_j')||_2$, where $|| \cdot ||$ denotes L2-norm. Also, $\mathcal{L}_{align_1}$ can be extended to a supervised version considering the LR masks are available, which are finally $\mathcal{L}_{align_1^*} = \mathcal{L}_{align_1} + \lambda_2 \mathcal{L}_{sim}(Y_j', Y_i \phi_{i \to j})$. The results of these two versions of registration loss had no significant differences based on our experiments.

## 2.3. Intertwined-tuning segmentation and registration

After pre-training the segmentation $\mathcal{S}_1$ and $\mathcal{G}_1$, we fine-tune them by intertwining the segmentation and registration steps. Similarly to the previous stage, we randomly select two different views from a single patient at one time, $I_i', I_j' \in I_1', I_2', I_3'$. We also introduce two new loss functions, cross-view supervision, $L_{cons_1}$ and $L_{cons_2}$ as

$$\begin{aligned} \mathcal{L}_{cons_1} &= ||M_{Fj \to i} - M_i||_2 + ||M_{Fi \to j} - M_j||_2, \\ \mathcal{L}_{cons_2} &= ||F_{Mi} - M_i||_2 + ||F_{Mj} - M_j||_2, \end{aligned} \tag{2}$$

where $M_{Fj \to i} = \mathcal{S}(f_{j \to i})$ represents the segmentation masks obtained from the aligned image $f_{j \to i}$, and similarly for $M_{Fi \to j} = \mathcal{S}(f_{i \to j})$. $F_{Mi} = \mathcal{S}(I_i') \circ \phi_{i \to j}$ was obtained by first segmenting the original image $I_i'$ and then registering the segmented mask into the pose of image $I_j'$. Similarly for $F_{Mj} = \mathcal{S}(I_j') \circ \phi_{j \to i}$. $\mathcal{L}_{cons_1}$ encourages the segmentation network $\mathcal{S}$ to predict masks from a single view of the image that looks similar to those from aligned images, and $\mathcal{L}_{cons_2}$ encourages the registration network $\mathcal{G}$ to align the segmented masks to have more overlap. These two intertwined regularizations are applied separately to $\mathcal{S}$ and $\mathcal{G}$ to achieve more accurate and precise segmentation and registration.

In training stage 2, the AlignNet and Segmentor are updated iteratively, and the final objectives to update the segmentation network and registration network are $\mathcal{L}_{seg_2} = L_{seg_1} + \alpha_1 L_{cons_1}$, $\mathcal{L}_{align_2} = L_{align_1} + \alpha_2 L_{cons_2}$, respectively.

## 2.4. 1-D Gaussian fusion to generate high-resolution mask

During inference, we take the three views of scans $I_1'$, $I_2'$ and $I_3'$, and align the last two views ($I_2'$ $I_3'$ into the axial scans $I_1'$) first; then these aligned scans are fed into the segmentation networks to get their predicted masks. Without loss of generality, let us examine the sagittal view as an example. We introduce a sampling matrix called $S_3$, that reflects the sampling

voxels in the up-sampled image matrix, where $S_3[i, j, k] = 1$ when this voxel point is taken from the original LR image $I_3$ instead of an interpolated value. Thus we can see that the matrix $S_3$ is like a matrix with the YZ-plane assigned to 1 at regular intervals in the x-dimension, as shown in Figure 2 (bottom branch). Inspired by descriptions of uncertainties in MRIs (Van Reeth et al., 2012), we found that 1D-Gaussian was a perfect match with the uncertainties between slices. Thus, we apply a 1D-gaussian along the through-plane dimension (x dimension for sagittal views) with a Gaussian kernel size of $d_s/2$, we could get a probability matrix $(P_3)$ that reflects the confidence of this scan at each voxel, and this probability matrix can be aligned into the registered images as $P_{3 \to 1} = S_3 \circ \phi_{3 \to 1}$. The final mask is fused by the following equation:

$$M_{hr} = \frac{1}{\omega_1} M_1 * P_1 + \frac{1}{\omega_2} M_{F2 \to 1} * P_{F2 \to 1} + \frac{1}{\omega_3} M_{F3 \to 1} * P_{F3 \to 1}; \qquad (3)$$

where $w_1$, $w_2$, $w_3$ are the normalization factors with $\omega_i = \frac{P_i}{\sum_1^3 P_k}$.

## 3. Experiments

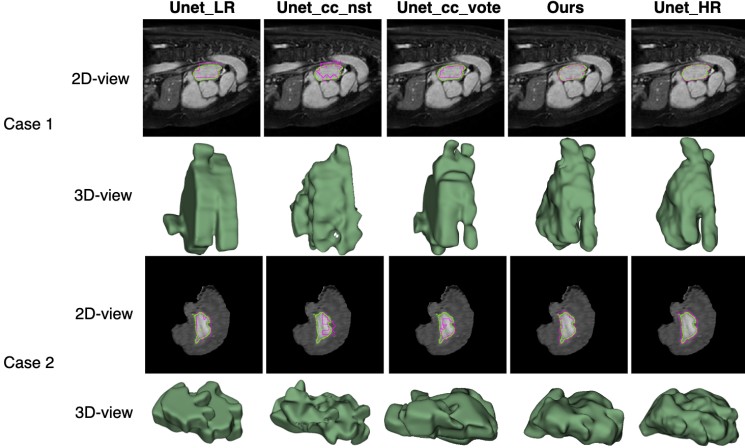

Figure 3: Qualitative results of the high-resolution mask generation methods. The methods depicted here are Unet-LR, Unet-cc-nst, Unet-cc-vote, Ours, and Unet-HR. Here are two examples from *Heart 16* (upper two rows) and *Brain 16* (bottom two rows), respectively. For each example, the first row is the 2D-slice view of the HR images with masks; the green curves are the ground truths, and the pink curves are the predictions. The bottom row shows the 3D view of the objects.

### 3.1. Datasets

As above-mentioned, most body parts are not imaged in high-resolution (HR) MRI, so to prove our method's effectiveness, we test our method by the body parts with HR MRI

available. By training the networks on the LR images, we can evaluate our method's performance on the HR masks. In this work, we obtain two MRI datasets, *Brats* (Menze et al., 2015) and *Heart* (Simpson et al., 2019).

*Brats* contains 273 patient studies. We split them into 164 studies in the training set and 42 in the test set. The original volumes are isotropic, with a voxel spacing of 1 mm * 1 mm * 1 mm. *Heart* contains 20 patient studies We split them into 15 studies in the training set and 5 in the test set. The original images have a voxel spacing of 1.25mm*1.25mm*1.37mm, which has a slightly lower but acceptable resolution in the third dimension. To make a consistent 3D input size for networks, we first resample the volumes with a 1mm*1mm*1mm voxel spacing. For each of the datasets, we start by centering the object and padding it to a cube of $256 \times 256 \times 256$ voxels. Then, we apply a random 3D affine transformation, consisting of a rotation ranging from -45 degrees to 45 degrees and a translation ranging from -30 voxels to 30 voxels, and auto-contrast to imitate the MRIs gotten at different times. To get the LR images, we resample each dimension with a slice distance of 8 mm (noted as datasets *Brain 8* and *Heart 8*) and 16 mm (noted as datasets *Brain 16* and *Heart 16*), where the resampled images have dimensions 32 for voxel spacing of 8 mm and 16 for voxel spacing of 16 mm.

## 3.2. Baseline methods

We compare our methods with the following: The first baseline, *Unet-LR*, is a segmentation algorithm (3D Unet) that uses all low-resolution images with all available views, similar to our stage 1 segmentation ($\mathcal{S}_1$). Second, *Unet-cc-img-nst*, is the Unet-LR plus image correlation, a traditional, non-learning-based iterative registration method that uses image correlation as a similarity metric and has a convergence minimum value of 1E-6 to register three-view images. Then the segmented masks are fused by a 3D nearest neighbor interpolation (Unet-cc-img-nst). Third, *Unet-cc-img-vote*, a similar version of the second but using majority voting, commonly used in ensemble learning, as the final fusion algorithm. The fourth and fifth, *Unet-cc-msk-nst* and *Unet-cc-msk-vote*, are similar versions of the second and third but apply registration directly to the segmented masks. Lastly, *Unet-HR* is trained on 1 mm* 1 mm* 1 mm voxel-spaced HR images. It is not appropriate to compare to the other methods, but it can serve as an "upper bound" for all methods.

## 3.3. Metrics

We compare the predicted masks with the ground-truth HR masks $Y_{HR}$. To make the comparison fair, we assume that the HR masks are perfectly aligned with the axial view images, as the axial view is unchanged. By directly calculating the metrics on the predicted axial view masks, we can compare the predicted LR masks $M_1$ with HR ground truth masks $Y_{HR}$. We quantify the generated HR masks with $M_{HR}$ by first the dice score (DSC) which quantifies the overlap of the predicted masks and HR ground truths. Then, the under segmentation (US), over segmentation (OS), root mean squared (RMS) (Monteiro and Campilho, 2006) which quantify the precision of the predicted masks. Lastly, in light of the fact that the DSC evaluates overlap but places little emphasis on the shape information of segmented volumes, we also add a boundary accuracy (mBA) (Taha and Hanbury, 2015) that focuses on the shape details.

### 3.4. Implementation details

For our methods, during training stage 1, Segmentor and Alignet are trained separately with a learning rate of 0.001 for 40 epochs. During training stage 2, the learning rate is set at 0.0001 with a step-wise learning rate decay for every 200 iterations as gamma $= 0.9$, and stage 2 training is done for 60 epochs. Before feeding into the networks, rescale intensity after a 2%-98% thresholding was applied to normalize image intensities into [0,1]. During all the training, the batch size is set to 2, and 2 GPUs with the Geforce A6000 are used. For the hyper-parameters, $\alpha 1 = \alpha 2 = 0.05$, $\lambda_1 = 0.1$, and $\lambda_2 = 0.05$, and it was chosen by the experiments on *Heart 16* dataset among several hyper-parameters combinations. All the baseline methods are set for training over 100 epochs with the same learning rate decay and batch size as our method.

Table 1: Quantitative results of the high-resolution mask generation methods. Dataset 8 and 16 note the slice distance in the through-plane direction. Results are reported based on 4 repeated training sections with average and standard deviation ($\pm$). our-1seq denotes the evaluation situation when only 1 view (axial-view) is available, and our-2seq denotes 2 views (axial-view and sagittal-view) are available.

| Methods | Heart 8 | | | | | Brain 8 | | | | |
|---|---|---|---|---|---|---|---|---|---|---|
| | DSC | US | OS | RMS | mBA | DSC | US | OS | RMS | mBA |
| Unet-LR | 0.863±.008 | 0.118±.015 | 0.151±.013 | 0.141±.015 | 0.850±.010 | 0.839±.011 | 0.127±.013 | 0.173±.010 | 0.166±.013 | 0.819±.012 |
| Unet-img-cc-nst | 0.819±.017 | 0.202±.019 | 0.156±.018 | 0.182±.019 | 0.803±.021 | 0.634±.024 | 0.194±.022 | 0.397±.041 | 0.195±.039 | 0.789±.029 |
| Unet-img-cc-vote | 0.879±.014 | **0.063**±.012 | 0.174±.015 | 0.130±.013 | 0.859±.016 | 0.813±.019 | 0.058±.020 | 0.262±.032 | 0.196±.033 | 0.807±.019 |
| Unet-msk-cc-nst | 0.799±.026 | 0.121±.028 | 0.251±.023 | 0.204±.028 | 0.814±.022 | 0.728±.036 | 0.086±.033 | 0.359±.039 | 0.279±.039 | 0.770±.035 |
| Unet-msk-cc-vote | 0.863±.022 | 0.198±.028 | **0.064**±.025 | 0.178±.024 | 0.842±.019 | 0.761±.029 | **0.053**±.032 | 0.330±.033 | 0.216±.021 | 0.776±.027 |
| Ours-1seq | 0.871±.005 | 0.098±.012 | 0.117±.010 | 0.108±.012 | 0.855±.009 | 0.849±.007 | 0.142±.008 | 0.151±.011 | 0.157±.009 | 0.821±.003 |
| Ours-2seq | 0.886±.006 | 0.095±.015 | 0.121±.016 | 0.114±.015 | 0.857±.011 | 0.855±.011 | 0.136±.009 | 0.131±.015 | 0.134±.011 | 0.830±.007 |
| Ours(SuperMask) | **0.896**±.002 | 0.093±.010 | 0.113±.010 | **0.105**±.010 | **0.870**±.009 | **0.883**±.012 | 0.109±.014 | **0.116**±.015 | **0.126**±.015 | **0.848**±.010 |
| | Heart 16 | | | | | Brain 16 | | | | |
| | DSC | US | OS | RMS | mBA | DSC | US | OS | RMS | mBA |
| Unet-LR | 0.812±.009 | 0.201±.018 | 0.173±.016 | 0.189±.018 | 0.780±.014 | 0.81±.014 | 0.134±.021 | 0.224±.024 | 0.195±.025 | 0.797±.019 |
| Unet-img-cc-nst | 0.664±.054 | 0.338±.065 | 0.330±.045 | 0.336±.063 | 0.687±.035 | 0.638±.099 | 0.441±.097 | 0.162±.056 | 0.347±.095 | 0.794±.074 |
| Unet-img-cc-vote | 0.683±.035 | 0.319±.066 | 0.302±.055 | 0.310±.063 | 0.703±.045 | 0.638±.095 | 0.504±.094 | **0.042**±.007 | 0.359±.094 | 0.715±.029 |
| Unet-msk-cc-nst | 0.683±.043 | 0.176±.054 | 0.401±.074 | 0.312±.064 | 0.730±.043 | 0.629±.104 | 0.464±.122 | 0.125±.067 | 0.353±.069 | 0.719±.054 |
| Unet-msk-cc-vote | 0.763±.024 | **0.059**±.007 | 0.357±.045 | 0.256±.043 | 0.766±.016 | 0.670±.047 | **0.040**±.005 | 0.460±.073 | 0.321±.071 | 0.728±.043 |
| our-1seq | 0.819±.008 | 0.200±.016 | 0.157±.012 | 0.182±.015 | 0.791±.016 | 0.820±.013 | 0.156±.024 | 0.190±.022 | 0.184±.023 | 0.801±.017 |
| our-2seq | 0.833±.005 | 0.187±.014 | 0.146±.015 | 0.177±.014 | 0.798±.015 | 0.836±.013 | 0.140±.027 | 0.185±.015 | 0.179±.017 | 0.810 ±.011 |
| Ours(SuperMask) | **0.875**±.004 | 0.139±.012 | **0.131**±.014 | **0.137**±.013 | **0.842**±.010 | **0.861**±.015 | 0.108±.017 | 0.165±.020 | **0.152**±.017 | **0.823**±.014 |
| | HR (upper bound) | | | | | HR (upper bound) | | | | |
| Unet-HR | 0.923±.009 | 0.102±.010 | 0.051±.006 | 0.082±.009 | 0.908±.013 | 0.900±.014 | 0.068±.013 | 0.117±.017 | 0.110±.015 | 0.877±.016 |

### 3.5. Results and discussion

Table 1 shows the results for the HR mask generation on various datasets. It is shown that our method outperforms all the baselines on all the datasets when only LR images are available. Compared with the drop in performance from Unet-HR to Unet-LR, there is a drop in DSC of 0.06 for slice distances of 8 mm and a drop of 0.112 and 0.09 for 16 mm. We can see that only training a simple 3D segmentation network with LR images and masks is not enough to achieve acceptable HR masks. The loss of information between through-plane slices is detrimental. The more distant the slices are from each other, the lower the resolution of the image is, and the more significant this performance reduction is. This could also be seen in Figure 3, where, despite up-sampling to the real-world coordination, the segmented masks from Unet-LR have a very distinct ladder-like appearance.

Even when only getting LR images, our method gets an increase in DSC of 0.03 and 0.043 for a slice thickness of 8 mm and a significant increase of 0.052 and 0.09 for a slice thickness of 16 mm, for the heart and brain datasets, compared with Unet-LR. This confirms the ability and effectiveness of our method to combine more details by extracting information from non-orthogonal LR images. Comparing RMS, it can be reduced by about 28% on the *Heart 16* dataset and 22.1% on the *brain 16* dataset. Our approach can substantially reduce the RMS due to the through-plane uncertainty. Using mBA, we observe an obvious increase in mBA for our methods, especially with a slice thickness of 16 mm, where a large part of the boundary details is lost from the input images. Although there are some smaller US for nearest or voting fusion methods, they suffer from a much larger OS, indicating an over-segmentation bias. Our method achieves a better balance.

The drop in performance for baselines Unet-img-cc-* and Unet-msk-cc-* implies that generating HR masks from multiple views requires highly precise image registration. The traditional image registration methods based on image correlation can not handle the low through-plane resolution well at different directions; see Appendix B for more details. Using failed image registration to construct segmentations from multiple views is detrimental. Our learning-based registration, combined with the target object's segmentation, can circumvent this problem and make connections between views, which contributes to the fusion of the target object's segmentation. Also, because the training was a two-view setting, our work can be easily extended when only two views are available (our-2seq), and also get an improvement. More details of the effectiveness of each component are shown in appendix D Table 4. Also seen from the illustration in Figure 3, our final generated masks, though free from any HR images or masks during the training and evaluation, can preserve the details of the object boundaries and have a high consensus with the HR masks.

## 4. Conclusion

In this work, we proposed a method *SuperMask* that generates high-resolution target segmentation from multiple unaligned 2D MRIs. We experimentally demonstrate the effectiveness of our methods for improving segmentation performance when only low-resolution masks are available. Importantly, our method does not require HR images in the training stage, which makes it broadly applicable. Future work will consider extending the method from generating HR masks to generating HR images.

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

## Appendix A. Illustration of the orientation of three different volumes

Figure 4 shows that the ideal assumption for multi-view MRIs is that axial/sagittal/coronal views are orthogonal to each other with no translation offset. However, in reality, these three perspectives are based on an anatomical term in which the objects may have large positional movements or orientations. If we translate these three views into real-world units and directly combine these three scans (Figure 4 (c), (d), (e)), the ulna/radius bones cannot overlap due to the positional/orientation alterations.

## Appendix B. The results for image registration

Figure 5 shows that, when we are to align coronal views and sagittal views which have the low resolution at different planes, traditional registration can achieve to-some-extent the correction between images. But the accuracy of its correction is not as high as that of our method. For example, if we look at the bottom two rows in Figure 5 (aligned sagittal for trade-img), it has a rotation error with the other two views after registration. These slight imperfections are detrimental if we want to rely on the predicted mask on these aligned images.

It can also be seen from Table 2, using our 2-stage intertwined learning can guarantee better agreements between views, where the DSC between views is higher compared with an isolated set of image segmentation and registration (Unet-img-cc-*). Also, applying the registration directly to segmented masks can achieve a large overlap, but these alignments might be far from the real objects (i&HR are lower).

## Appendix C. Details of hyperparameter selection

We selected our model's final hyper-parameters based on the evaluated DSC on the *Heart 16* dataset, where we put extra 4 volumes in training set as the evaluation set here. There are enormous combinations of parameters $\alpha_1$, $\alpha_2$, $\lambda_1$, $\lambda_2$, and we did not through all the

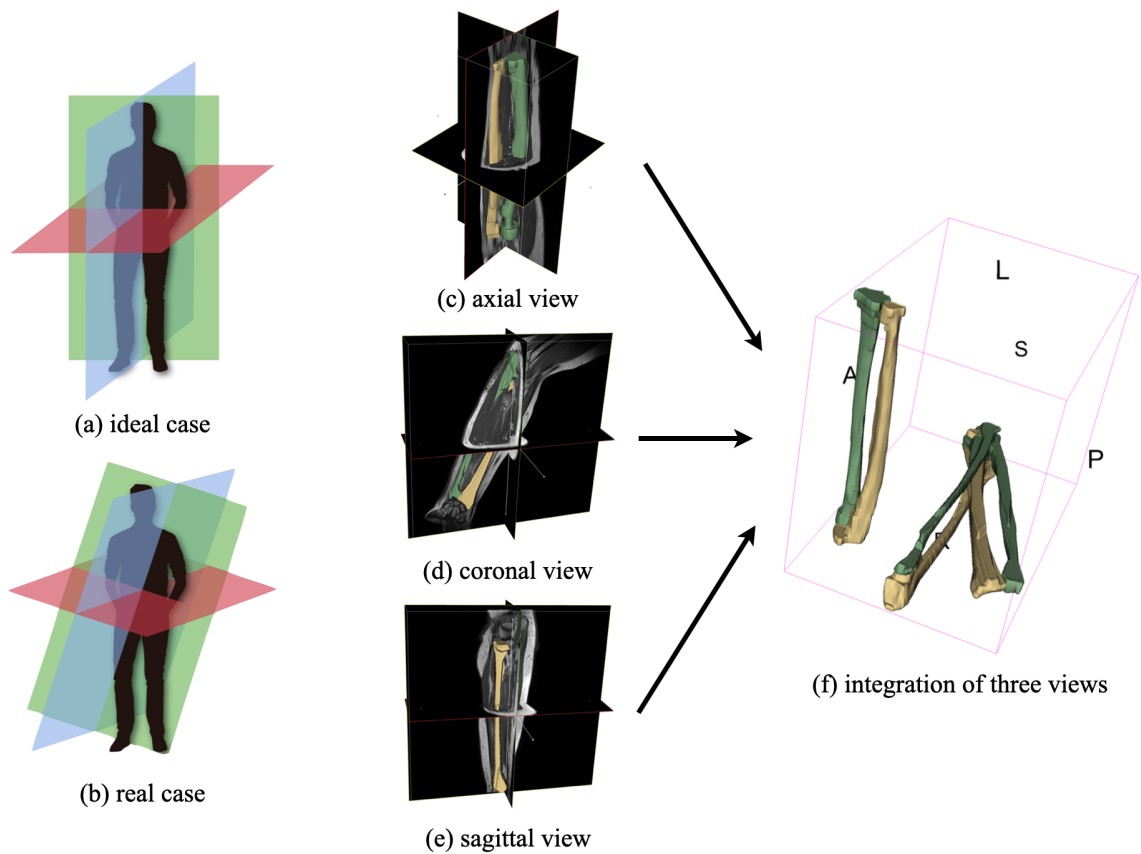

Figure 4: Illustration of input MRIs and the orientation of three different volumes. (a)(ideal case) anatomical coordinate system aligned with image coordinate system; (b)(real case) anatomical coordinate system not aligned with image coordinate system; (c)axial-view scans; (d)coronal-view scans; (e)sagittal-view scans; (f) integration of three views from the same patient in real-clinical forearm MRIs.

Table 2: Quantitative performance of image registration for multi-view MRIs. Shown are the DSC between segmented masks on the aligned views of the Brat 16 dataset. 1&2 means DSC calculated on view 1 and view 2 masks.

| DSC between views | 1&2 | 1&3 | 2&3 | 2&HR | 3&HR |
|---|---|---|---|---|---|
| Unet-img-cc-* | 0.71 | 0.661 | 0.697 | 0.765 | 0.717 |
| Unet-msk-cc-* | 0.750 | 0.725 | 0.696 | 0.749 | 0.715 |
| Ours | 0.784 | 0.75 | 0.755 | 0.816 | 0.783 |

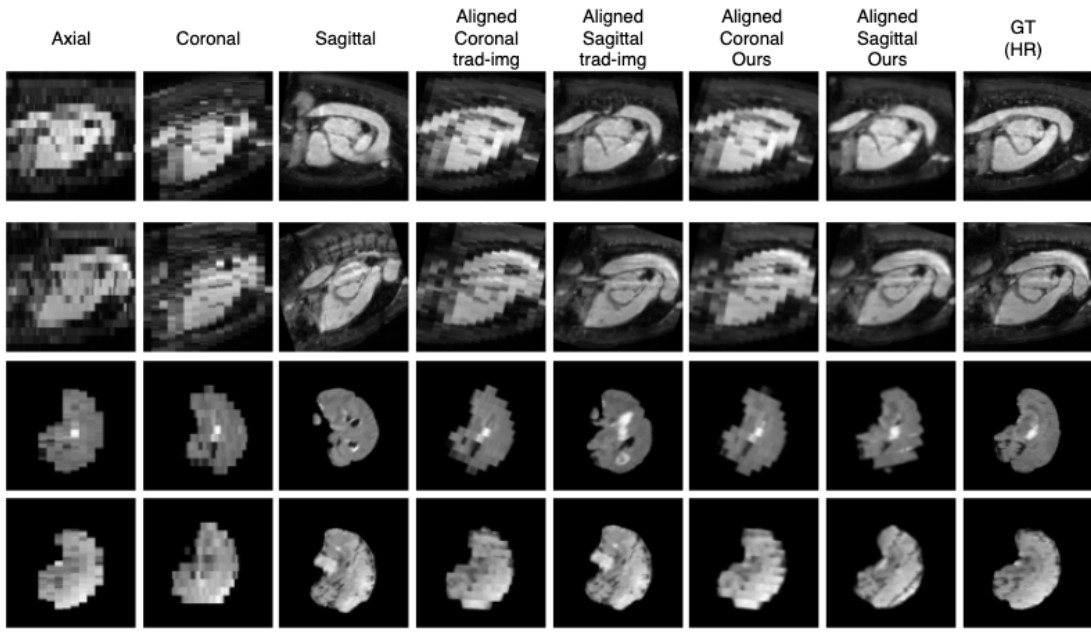

Figure 5: Qualitative results for image registration from multi-views,

potential parameter settings. Based on the five different combinations, we found that $\lambda_1$, $\lambda_2$, which controls the registration loss, were preferable under a scale of 0.5. $\alpha_1$, $\alpha_2$, which controls the scales of regularization terms, are not too influential on the final performance. We assumed that is because of the two-stage training, the previous loss terms as $L_{align1}$ and $L_{seg1}$ were already optimized in pre-training, and stage 2 would care more from the gradients contributed from $L_{cons1}$ and $L_{cons2}$ no matter the scaling is. We agreed that we are not thoroughly optimizing these parameters on each individual dataset, but the general selection rule for the scales of parameters found by these five combinations could give us a relatively good performance on other datasets.

Table 3: The details of hyper-parameter selection on *Heart 16* dataset, and the supporting metric is DSC.

| model version | $\alpha_1$ | $\alpha_2$ | $\lambda_1$ | $\lambda_2$ | DSC |
|---|---|---|---|---|---|
| model 1 | 0.1 | 0.1 | 0.1 | 0.1 | 0.87 |
| model 2 | 0.2 | 0.2 | 0.5 | 0.5 | 0.845 |
| model 3 | 0.05 | 0.05 | 0.1 | 0.05 | 0.871 |
| model 4 | 0.5 | 0.5 | 0.1 | 0.05 | 0.867 |
| model 5 | 0.05 | 0.05 | 0.1 | 0.05 | 0.872 |

## Appendix D. The results for ablation studies

This section shows the ablation studies for our method that demonstrate the effectiveness of each component of our method. *Our-1-stage* is our methods' one-stage training without an intertwined regularization. *Our-voting* is replacing the 1D-fusion in our method with voting, and *Our-voting* is replacing the 1D-fusion in our method with the nearest interpolation.

Table 4: Ablation study of each component of our model.

| Ablation methods | componenets | | | heart-16 | | | | |
|---|---|---|---|---|---|---|---|---|
| | two-stage | Alignet | fusion | DSC | US | OS | RMS | mBA |
| Our-1-stage | 0 | 1 | our fusion | 0.849 | 0.178 | 0.120 | 0.154 | 0.819 |
| Our - voting | 1 | 1 | voting | 0.826 | 0.154 | 0.161 | 0.157 | 0.821 |
| Our - nearest | 1 | 1 | nearest | 0.763 | 0.286 | 0.227 | 0.237 | 0.750 |

