# OpenReview forum: "SuperMask: Generating High-resolution object masks from multi-view, unaligned low-resolution MRIs"
_MIDL.io/2023/Conference — MIDL 2023 Oral_

### Official Review · Reviewer_vkMg · 2023-01-26

**Confidence:** 3
**Preliminary Rating:** 4
**Recommendation:** Poster

**Summary:**

The method presented aims to generate 3D high-resolution images from 2D low-res to enhance segmentation on MRI acquired with three different views not necessarily orthogonal or well-aligned.










































































































...





**Strengths:**

* exciting combination of registration and segmentation with superresolution.
* high-resolution images seem no the be needed during training, which makes the approach particularly appealing.








.

**Weaknesses:**

* The method is multi-step, which makes it convoluted and requires pretraining.
* No clear definition of learning paradigm has been presented -->, e.g., supervised, unsupervised, and so on...
* The data used are simulated low-resolution data, so they are synthetic. The data distribution is not necessarily comparable to directly acquired low-resolution data. In this sense, the impact and relevance of this work within a clinical environment cannot be fully understood with the experiments proposed.
* The baselines chosen are currently too many, making it not readily apparent which is the ground truth to compare.
* There is no mention of hyperparameter optimization or a criterion adopted to perform it, which makes me wonder whether the results obtained are optimal.
* The comparison of results lacks statistical analysis, the authors are only comparing the mean values, but this may be meaningless when considering std and null-hypothesis tests.

**Deanonymize Review:**

no

**Detailed Comments:**

Major Comments
a) The first paragraph of the intro is deceiving, especially concerning the clinical problem definition. The authors have made some assumptions, but they must be clearly stated. Here is an example of the interpretation I gave by reading the paragraph:
	1. only 3D MRI segmentation is important, if 2D or another image modality not necessarily;
	2. all MRI acquisitions have a low-resolution acquisition -> this is false. One could acquire high resolution, of course, paying the price in terms of acquisition time;
	3. all the MRI acquisitions have an in-plane direction -> this is not true. See below;
The suggestion is to revise the whole first paragraph double, checking whether its interpretation is in line with what the authors desire to achieve;
b) The authors improperly refer to an in-plane/out-of-plane direction. This is quite improper for MRI, given that it better speaks of read-out and phase encoding directions. It may be beneficial to revise how to address the direction throughout the paper.
c) It may be good to specify from the introduction that low-resolution is intended with low-resolution acquisition along one spatial dimension. Still, the method requires having this low-resolution acquisition in three planes (not necessarily orthogonal, but they need to be non-aligned). This point currently needs to be clearly described/written. A significant textual revision of the intro could address this point. In this sense, revise the start of the method, there the goal needs to be clearly described since it needs to be clarified that the methods start from 2D acquisition to full 3D ones.
d) Throughout the paper, which paradigm is adopted to perform learning is not mentioned, e.g., supervised, unsupervised, self-supervised, and so on (it may be a hybrid, depending on the step considered). It should be readily made clear in the intro/methods.
e) The expression "intermingled/intertwined" learning needs to be more readily evident; probably good to anticipate/add what the authors mean the first time it appears.
f) In the introduction (towards the end of the first page), the requirement of well-aligned and near orthogonal refers to Askin Incebacak et al., 2022. In principle, well aligned depends on the learning paradigm chosen, e.g., for supervised learning. This is undoubtedly the case. What concern the near orthogonality, this is seldom a hard constraint from the proposed methods. For example, even in the paper cited (Askin Incebacak et al., 2022), orthogonal views are used because available but never mentioned are strictly necessary. I suggest revising this point here.
g) The heart dataset is resampled going superresolution, which may add some blurriness + if the misalignment of the slice was present in the original slices, by resampling (also when considering the downsampling to 8 and 16), the 2D misalignment might be lost. Is the dataset used correctly or properly to assess non-aligned sets?
h) Too many methods are considered baseline. It may be more evident to choose only a couple as a baseline and experiment on the impact of the loss function used.
i) The results highlight the effectiveness of extracting information on non-orthogonal slices. However, the method needs a clear description of the non-orthogonal LR images; otherwise, it seems this point is not strongly supported.

Minor Comments
The first sentence of the body: not only is 3D segmentation important, but it is probably better to omit 3D.
Page 2 "achieve a more optimal registration" --> English is not correct = being optimal is already a superlative. There is no such thing as being more than optimal.
2.1 The prime symbol applied, e.g., to I1, I2, and so on (I1', I2',...), is not well introduced. Textually explaining its meaning could clarify.

**Paper Type:**

both

**Questions To Address In The Rebuttal:**

a) better description of the rational/clinical problem;
b) revise the definition of the method;
c) reconsider what is a baseline and what can be placed as an experiment;
d) strengthen statistical analysis;
e) comment on hyperparameter optimization;
f) clarify the in-plane/out-of-plane for MRI and that LR is generally 2D

---

### Official Review · Reviewer_awAy · 2023-01-31

**Confidence:** 4
**Preliminary Rating:** 4
**Recommendation:** Poster

**Summary:**

This work proposes a method to combine multiple anisotropic MRI scans to generate a high-resolution segmentation using a combination of deep learning-based registration and segmentation. The method is trained and evaluated using high-resolution (HR) images from BRATS (273 patients) and HEART (20 patients) that are resampled and augmented by the authors to generate low-resolution (LR) images for training.

**Strengths:**

- Generated HR segmented volumes seem accurate
- No HR imaging data required for training
- Good benchmarks are provided for evaluation: two 3D Unets for image segmentation, combined with a nearest neighbor or majority voting interpolation to combine the different segmentation masks, and two Unets that use registration to combine the masks. The authors also provide the results of a Unet trained on HR images for comparison.


**Weaknesses:**

- The proposed method is only evaluated on data that are artificially downsampled and augmented. It remains unclear to what extent this method would work on real LR data acquired in clinical settings.
- The applicability of the proposed method might be limited because it is designed for a rather specific use case, namely when multiple (>3) MR scans of the same anatomy are available, each with one dimension scanned at a high resolution.
- More background information would be helpful: are there any other methods available that are capable of combining segmentations from multiple MRI scans?
- Although this work uses a smart combination of (deep learning-based) segmentation and registration techniques, I’m not sure about the novelty of the individual elements, particularly the proposed “intertwined” regularizations and the 1-D Gaussian fusion.


**Deanonymize Review:**

no

**Paper Type:**

methodological development

**Questions To Address In The Rebuttal:**

- Elaborate on the applicability of the proposed method on real low-resolution MR data. Do the authors have access to such data? It would be great to test the SuperMask method on such a dataset.
- Can the authors comment on the clinical problem/use case and the problem they are solving in this work? Is it common to have multiple (> 3) MR scans of the same anatomy available, each with one dimension scanned at a high resolution? How realistic is this situation and how often does it actually occur? And to what extent would the performance of the method be affected when the different MRI scans have very different image contrasts (making deep learning based segmentation more challenging), or if one dimension of the image is not available at a high resolution?
- Provide background information on alternative methods capable of fusing segmentations of multiple MR scans (if any).

---

### Official Review · Reviewer_1Xn5 · 2023-02-02

**Confidence:** 3
**Preliminary Rating:** 4
**Recommendation:** Poster

**Summary:**

This method proposes SuperMask that generates high-resolution target segmentation from multiple unaligned low-resolution MRIs. The work demonstrates the effectiveness of SuperMask for improving segmentation performance when only low-resolution masks are available, without requiring HR images or direct 3D masks in the training process.


**Strengths:**

This work considers misalignment between various orientations of a 3D anatomical scan when using multiple-oriented images to generate high-resolution masks.  The target images considered for computing masks suffer from larger displacements and different dimensions and hence traditional image registration methods may not work as desired, hence deep learning registration technique is used as a core step in the overall mask prediction pipeline.

The method does not use high-resolution images or masks directly in the training process.


**Weaknesses:**

The authors mention two regularisation terms in the last paragraph of the introduction which talks about the core contributions of the work. It is not clear what the contributions of these regularization terms are. This information is also missing in the abstract.

A suddenly introduced notation Ri’ is not defined in equation 1. The authors need to define Ri as it is the critical section of the manuscript. The sentence above equation 1 says $L_{sim}$ penalizes the difference between moving image and fixed image but in equation 1, the moved image $f_{i→j}$ is given. The authors must explain this part clearly by supplementing notations along with names. From Figure 2, I found that $L_{sim}$ is that loss between the moved image $f_{i→j}$ and the fixed image I’j. Similarly Ri’ is introduced in Section 2.3 also. The authors need to ensure consistency in notations.

This work still needs three orientations and the corresponding ground truth masks for the overall training pipeline.

The authors claim that 1D Gaussian fusion is novel, however, the motivation behind this approach is not discussed.


**Deanonymize Review:**

no

**Detailed Comments:**

In the Figure 4 caption, while the first two sub-figures explain the alignment between the human coordinate system and imaging coordinate system, the name of the anatomy of the third, fourth, and fifth sub-figures are missing.

In Figure 2, Training phase 2, the meaning of the word “interwine” is not clear.

The word “novel” for 1-D Gaussian fusion seems to be used in multiple places, which, in my opinion, is unnecessary. In fact, the word, “novel” is used in four places in the whole manuscript..!

Figure 2 fonts are extremely small, making reading on a printout (100% zoom level) very difficult. The authors must consider increasing the font size to improve readability.

The pink and green curves are not clear in Figure 3, the authors must highlight them with thick curves.


**Paper Type:**

both

**Questions To Address In The Rebuttal:**

How is P1 calculated in equation 3? Meaning, which orientation is out of plane for axial slices?

What is R1 in equations 1 and Section 2.3?

What is ds in Section 2.4?

How does the probability matrix reflect the confidence of this scan at each voxel?

In Table 1, Unet-msk-cc-vote gives  OS, the best value of 0.064 for Heart 8 is present, but authors have highlighted their method which gives the second best value of  0.113. Reason?

---

### Meta-Review · Area_Chair_hABn · 2023-02-21

**Recommendation:** Accept (Poster)
**Confidence:** 5

**Metareview:**

This is a very relevant submission with important practical applications, particularly since isotropic data are not required in training.
Some clarity issues were brought up by the reviewers, but these were successfully addressed / rebutted by the authors. Evaluation is thorough - even though evaluation with *acquired* rather than simulated low-resolution images would have been desirable. Lack of discussion of related methods and lack of statistical analyses were brought up by reviewers but successfully addressed in the rebuttal / revision.  Novelty is limited but this is overall a solid paper, particularly after the rebuttal.